# Neurophysiological Correlates of Top-Down Phonological and Semantic Influence during the Orthographic Processing of Novel Visual Word-Forms

**DOI:** 10.3390/brainsci10100717

**Published:** 2020-10-09

**Authors:** Beatriz Bermúdez-Margaretto, David Beltrán, Yury Shtyrov, Alberto Dominguez, Fernando Cuetos

**Affiliations:** 1Centre for Cognition and Decision Making, Institute for Cognitive Neuroscience, National Research University Higher School of Economics, 101000 Moscow, Russia; 2Instituto Universitario de Neurociencia (IUNE), Universidad de La Laguna, 38071 Tenerife, Spain; dbeltran@ull.edu.es (D.B.); adomin@ull.es (A.D.); 3Facultad de Psicología, Universidad de La Laguna, 38071 Tenerife, Spain; 4Institute for Clinical Medicine—Center for Functionally Integrative Neuroscience (CFIN), Aarhus University, 8000 Aarhus, Denmark; yury@cfin.au.dk; 5Facultad de Psicología, Universidad de Oviedo, 33001 Oviedo, Spain; fcuetos@uniovi.es

**Keywords:** ERPs, phonology, semantics, reading, top-down processing

## Abstract

The acquisition of new vocabulary is usually mediated by previous experience with language. In the visual domain, the representation of orthographically unfamiliar words at the phonological or conceptual levels may facilitate their orthographic learning. The neural correlates of this advantage were investigated by recording EEG activity during reading novel and familiar words across three different experiments (*n* = 22 each), manipulating the availability of previous knowledge on the novel written words. A different pattern of event-related potential (ERP) responses was found depending on the previous training, resembling cross-level top-down interactive effects during vocabulary acquisition. Thus, whereas previous phonological experience caused a modulation at the post-lexical stages of the visual recognition of novel written words (~520 ms), additional semantic training influenced their processing at a lexico-semantic stage (~320 ms). Moreover, early lexical differences (~180 ms) elicited in the absence of previous training did not emerge after both phonological and semantic training, reflecting similar orthographic processing and word-form access.

## 1. Introduction

Extensive vocabularies of tens of thousands of words and their continuous build-up throughout the entire life span are unique features of human linguistic communication. Yet, the neurocognitive mechanisms of word acquisition remain poorly understood. One important but unresolved issue concerns the interplay between different levels of linguistic processing—e.g., phonological, orthographic, semantic, syntactic—across spoken and visual language domains. According to some suggestions [1,2], the more levels at which a novel word gains activation there are, the more interactive its processing will be, contributing to its faster lexicalization. The lexicalization of novel word-forms refers to the development of fully fledged representations involving activations across different levels of analysis, which, crucially, show dynamic interactions with other word representations. This process is particularly important for the efficient use of new vocabulary [3,4]. Indeed, such cross-level network representation likely operates during the reading process, contributing to the parallel activation of various information types, and hence to the efficient whole-form recognition [2,5,6]. In contrast, when the activation and mapping across different levels of representation is reduced or impossible due to the lack of experience at one or more levels, the word processing likely becomes more effortful, slower, and possibly more serial in nature. Therefore, previous experience with words at various processing levels likely leads to a more rapid and interactive processing for these stimuli, which in turn facilitates their lexicalization and hence their efficient use. The present study is aimed to investigate the neural dynamics of such cross-level interactivity during the acquisition of new vocabulary.

Several cognitive models of word recognition consider the interaction across multiple levels of analysis, in both visual [7,8,9] and spoken domains [10,11]. Moreover, such approaches imply a continuous bi-directional (bottom-up, top-down) flow of information, in which sensory input and prior knowledge interact to ensure the most efficient word processing [10,12,13]. As a result, one could expect the facilitation in orthographic analysis of novel written word-forms to a higher degree, when representations can be also activated at other, non-orthographic levels—i.e., phonological and semantic ones. This is indeed a very common situation when facing novel vocabulary in childhood and also during foreign linguistic immersion in adulthood. Very often, the orthographic form of a novel word is not accessible, as it has never been experienced visually, but its pronunciation or even its meaning can be accessed, as it has already been experienced outside of a reading context at a phonological and/or semantic level. Given the dynamic nature of visual word recognition, the activation of phonological and semantic information driven by top-down mechanisms likely facilitates the grapheme-to-phoneme decoding involved during orthographic processing; this would in turn contribute to a faster transition from serial letter-by-letter processing of novel written word-forms to a whole-word reading strategy, which is known to occur over the course of orthographic learning [14,15,16].

This question has been empirically addressed by prior behavioral research, showing facilitatory top-down processes from both the phonological and semantic levels in the orthographic processing of novel word-forms [5,17,18,19,20,21,22,23,24,25]. In general, the findings reported in these studies point out that phonological knowledge generally improves (e.g., speeds up) the orthographic processing of novel words. However, the evidence for a semantic benefit is more blurred and inconsistent; whereas recent studies inform of better reading performance after training word pronunciation in combination with meaning [17,22], other studies report no additional benefit from semantics [19,20,23], or claim that semantic facilitation depends on the depth of the orthographic system [5] or the specific script [25].

Although such studies have shed light on this question, other measures than behavioral are required to better determine the processes underlying the specific neurocognitive stages of visual word recognition and how they are influenced by previous training. To provide fine-grain information regarding the temporal dynamics of word processing, methods with a high temporal resolution are needed, such as electro- or magnetoencephalography (EEG, MEG). Indeed, several studies have used EEG/MEG methodology to isolate the specific processes taking place during various stages of visual word recognition, with particular event-related potential (ERP) components reflecting different mental operations. Thus, the brain responses elicited within first 200 ms of word processing have been related to orthographic analysis [26,27,28,29,30]. In particular, the amplitude of the P200 component has been suggested to reflect the differences in access to the orthographic form during visual word recognition. P200 is a fronto-centrally distributed component which usually shows larger responses for high- than for low-frequency words or pseudowords; it is sensitive to physical characteristics of the stimuli, as well as to their lexical frequency, and it is considered to index the holistic recognition of word-forms [31,32,33,34,35]. Later on, from ~250 to 500 ms, higher-level analyses are carried out, related to lexico-semantic access and contextual integration processes, as reflected in the amplitude of the N400 component; in particular, larger responses of this parietally distributed negativity are considered to reflect the difficulty in anticipating, processing, and integrating the word into the ongoing semantic context [36,37,38].

These ERP components, indexing processes carried out at different stages of visual word processing, also reflect changes in the visual recognition of words as a consequence of increased reading experience. Thus, several studies have reported facilitation in the processing of novel written words as a consequence of repeated visual exposure, indicating the top-down facilitation of processes within the orthographic level of representation. The majority of these studies have reported reductions in the amplitude of the N400 component elicited by novel written word-forms [39,40,41,42,43,44,45]. This reduction is considered to reflect the facilitated lexico-semantic access of novel written words due to repeated visual exposure under meaningful training contexts (i.e., in association with pictures or definitions, or embedding them within semantically constrained sentences). Moreover, the N400 reduction is often followed by the enhancement of a late positive component (LPC) associated with post-lexical processes. Specifically, such modulation is considered to reflect the formation and enhancement of episodic memory traces for novel written words across repeated exposure to them, contributing to their recognition [40,41,45,46,47]. Much less frequently found in these studies, however, is the modulation of earlier ERPs, reflecting a facilitation in processes related purely to the orthographic analysis of the novel written words. This could be due to the paradigms used in these studies, in which the task (i.e., semantic categorization, lexical decision) leads to a deeper processing of the novel written words, likely masking the modulation of earlier brain responses. In support of this argument, a recent study has reported the enhancement of early neuromagnetic responses (~100 ms post-onset) using a passive paradigm in which novel written words were repeated outside the participant’s focus of attention [48]. This early modulation was considered as an index for the fast and automatic formation of surface-form representations for novel written words.

Importantly, however, prior ERP research has not addressed the putative cross-level top-down facilitation during the orthographic processing of novel written words. Whereas some studies did report cross-level top-down facilitation during visual word recognition [49], it was only tested on familiar words, and thus cannot directly speak to the neurophysiological mechanisms underlying the cross-modal, top-down facilitation of the visual recognition of novel words. On the other hand, many other EEG/MEG studies have tested the effect of training new phonological word-forms, either with or without semantic information [50,51,52,53,54,55]. However, in these studies the effects of such phonological or phonological/semantic exposure with novel words were explored only within the spoken domain, showing, for instance, facilitation in the acquisition of novel spoken words when these stimuli resemble native phonotactic structures (and thus facilitated by pre-existing phonological representations) or lexical competition effects in the spoken recognition of familiar words (driven by the concurrent activation of the newly acquired ones). However, the intermodal effect that the phonological/semantic training (in spoken modality) could have on the orthographic processing of these stimuli (in visual modality) has not been neurophysiologically explored so far. Therefore, the neural mechanisms underpinning a putative facilitation in the visual recognition of word-forms driven by access to cross-modal levels of representation still remain poorly understood.

## 2. The Current Study

The present ERP study is aimed at evaluating the time course and functional role of cross-level top-down facilitation processes during the visual recognition of novel written word-forms. For this purpose, we carried out three different experiments in which we manipulated the training of novel words in their meaning and/or phonology. The impact of this training on the visual recognition of novel written word-forms was evaluated using a silent reading task in which the brain dynamics underlying the orthographic processing of the previously trained stimuli were measured and compared to those elicited by familiar, already lexicalized words. The choice of this task was motivated, as already reported in previous studies [42,48], by the shallowness of the processing it involves, likely more appropriate for the study of visual word recognition processes than other paradigms involving the categorization of the stimuli. Thus, in the first experiment the novel written words were presented in a silent reading task, but, crucially, no previous training was provided for these stimuli; therefore, this experiment served as a baseline. In the second experiment, however, the novel written words were trained in their phonology before their visual presentation at the silent reading task, and in the third one these stimuli were previously trained both in their phonology and meaning. Crucially, each experiment was conducted in a separate group of participants in order to detect the effects of each particular training and avoid any influence or carry-over of effects between the different training types. Importantly, the same materials and general procedures were used across the three experiments, as well as the same sequence of preprocessing steps and analysis of the EEG signal; this tight control over experimental condition enables the direct assessment of effects of the training manipulations rather than of the reading task per se.

We expected that the cross-modal top-down access to semantic and/or phonological codes during the reading of novel written words would cause a facilitation in their visual processing, leading to differential patterns of brain activation depending on the previous training. Thus, in experiment 1, we expected to find differences in the orthographic and lexico-semantic processing of familiar and unknown (non-trained) novel words; such differences at early and late stages of their lexical processing would be likely reflected in a lexicality effect in the P200 and N400 components. In experiment 2, the access to phonological codes, enabled by previous training, was expected to cause a facilitation in the phonological decoding of novel written words, which could, in turn, lead to a more automatized orthographic analysis, thus resulting in the reduction in the P200 lexicality effect; furthermore, the repeated exposure to novel words during the training might also cause an LPC modulation, as this late component has been found to be sensitive to stimulus repetition. Finally, the additional meaning-specific semantic training carried out in experiment 3 was expected to enable the recollection of a semantic reference during the reading of novel written words, thus likely affecting the amplitude of the N400 component.

## 3. Experiment 1: No Previous Training

### 3.1. Method

**Participants**: Twenty-two participants took part in the experiment for course credits (six males; mean of age = 21.2, SD = 2.43). All the participants gave their consent to take part in the experiment. The participants were native Spanish speakers, right-handed (mean score of 77.90 in the Edinburgh Handiness Inventory), and had normal audition and normal or corrected-to-normal vision. None of them reported a history of cognitive, neurological, or psychiatric disorders.

**Materials**. A total of 48 stimuli were selected for this experiment (see the footnote of Table 1). Twenty-four of them were unknown Spanish words (so-called obscure words, with a mean lexical frequency of 0 occurrences per million—i.e., *jínjol*, Eng. jujube, a type of buckthorn), used as novel words to for training. The participants were asked at the end of the experiment in order to ensure they were naïve about these stimuli. The other 24 stimuli were known, medium-frequency Spanish words, used as control stimuli (mean lexical frequency of 57.8 occurrences per million—i.e., balcón, Eng. balcony). All the stimuli were bi-syllabic, had a 5–6 letter length, had the same CVC (Consonant-Vowel-Consonant) structure, and were matched in different sublexical variables by means of the Buscapalabras Spanish database [56]—namely, in the number of letters and syllables, the number of orthographic neighbors, the bigram frequency, the mean syllable frequency, and syllabic structure (See Table 1).

Stimuli were presented during a silent reading task. In particular, the task consisted of two blocks of stimuli presented in a consecutive pseudo-randomized order—familiar words and novel word-forms. The novel and known words were separated into the different blocks to avoid the formation of associations between meaningful and meaningless items. Both sets of stimuli were presented only once during this reading task, with one trial per each of the stimuli. The sequence of stimulus presentation was randomized within each block.

**Procedure**. Participants were seated in a comfortable chair, in front of a computer screen where the stimuli would be displayed. After placing an EEG cap on the participant’s head (see details in the EEG recording and analysis section), the task was introduced. The participants were instructed to pay attention and to silently read the stimuli presented on the screen using sub-vocalic, covert articulation. The task started with six filler trials for demonstration. The stimuli were presented at the center of the screen in white 18-point bold Courier New font over a black background by means of the E-Prime 2.0 software (Psychology Software Tools Inc., Sharpsburg, PA, USA). Figure 1B illustrates the sequence of presentation during the reading task.

Therefore, no previous training was carried out in this experiment; it consisted of a single phase (testing) in which the participants were presented with novel and known words during the recording of their EEG activity (see Figure 1B). The aim of this first experiment was to stablish a baseline for the lexicality effect, and thus a comparison between the brain signals elicited during the reading of well-known and novel stimuli.

**EEG recoding and analysis**. During the task, EEG signals were recorded by 64 Ag/Cl active electrodes (actiCap, Brain Products GmbH, Gilching, Germany) and amplified and digitized with an ActiChAmp amplifier (Brain Products GmbH, Gilching, Germany) at a 1000 Hz sampling rate. Ocular activity was recorded by two electrodes placed at the horizontal and vertical canthus of the left eye. A reference electrode (Cz) was placed at the vertex. Two additional electrodes were placed at the left and right mastoids for offline reference. High and low pass filters at 0.1 and 100 Hz were applied during the recordings.

The preprocessing of the EEG data was carried out using the Brainstorm software [57]. A 30 Hz low-pass filter was applied and then the data were downsampled to 250 Hz and epoched between ~200 ms and 1000 ms post stimulus onset. The baseline was corrected using the 200 ms interval preceding the stimulus onset in each epoch. Independent component analysis (ICA) was used to remove ocular artifacts, and a triangular interpolation of bad channels was applied (average number of rejected ICA components: 2, range~=~0–4; average number of interpolated channels: 4, range~=~1–8). Additional artifact rejection (using exclusion criteria at ±100 µV) was applied to remove any remaining contaminated epochs (the mean number of rejected trials per condition was, for known words, 4, range~=~2–9, representing 17.99% of the data; for novel words, it was 3, range~=~1–7, 14.20% of the data). Data were re-referenced offline to average mastoid reference. Finally, the EEG epochs were averaged per subject and per condition (separately for known and for novel words) and the ERPs were computed (average number of included trials per condition: known words: 20, range~=~15–22; novel words: 21, range~=~17–23).

Statistical analyses of the ERPs were also carried out using Brainstorm. A permutation *t*-test was conducted across the whole ERP segment (time and space) in order to explore significant differences between known and novel words. A total of 1000 permutations were carried out for each sample point (300 time points by 60 channels = 18,000 sample points), in which the conditions (known and novel words) were contrasted by means of a paired *t*-test, thus allowing us to determine the exact time window and scalp topography of the differences between known and novel words. In order to correct for multiple comparisons and prevent false-positive rates, specific criteria were followed; in particular, only those differences maintained for a minimum of 5 consecutive samples (i.e., over 20 ms) in at least 3 adjacent channels and with alpha level of 0.025 (for two tailed-tests) were considered significant. Importantly, such number of samples ensured the detection of differences in early, orthographic-related ERP components, whose duration is usually short [26,27,28,29,30].

Furthermore, a source estimation analysis was carried out, in order to have a glimpse of the brain regions involved in the lexicality effect obtained at the surface level. For this purpose, the Minimum-Norm Estimate (MNE; [58]), implemented in the Brainstorm software, was used to obtain a current density map for each subject and condition (trained and novel words) at the time windows in which significant differences were obtained at the sensor level. These maps, representing the current density magnitudes (ampere per square millimeter), were calculated on a realistic head model (BEM) including 4025 nodes, defined in regular distances within the gray matter of a standard MRI (Montreal Neurological Institute’s average brain). Finally, the difference in current density map between novel and known words was subtracted. The choice of the MNE method was based on its higher sensitivity for superficial sources instead of for deep, intracranial generators, and its relative ability for source localization power with minimal a priori assumptions about the nature of the source [59].

### 3.2. Experiment 1 Results

A permutation *t*-test conducted for the comparison between known and novel untrained words showed significant differences (*p* < 0.025) in a period ranging approximately from 180 to 210 ms; no other difference was found at other time period across the whole ERP segment (see Appendix A). These differences were due to more positive activity for known than for novel words, distributed across the frontal and central scalp sites. Both the morphology of the ERP waveforms and the topography of the obtained difference suggested that this effect reflects the modulation of the P200 component. See Figure 2A.

Brain source estimation carried out in the 180–210 ms averaged time window revealed frontal and temporal areas of the left hemisphere, particularly at its anterior pole, as the most likely brain generators for the differences between known and unknown words found at the scalp surface level. In particular, the difference in the current density map between both conditions revealed a higher activation for known than for unknown words in the left superior, middle, and inferior temporal gyri, as well as at the left inferior frontal gyrus. Thus, the higher P200 amplitudes observed for known words in comparison to unknown word-forms were probably generated by a language-related perisylvian brain network in the left hemisphere. See Figure 2B.

### 3.3. Experiment 1 Discussion

In this experiment, brain activity differences between known and novel word-forms were observed in the amplitude of the P200 component, with larger responses for familiar than for novel stimuli. This P200 lexicality effect is similar to that found in previous studies [31,32,33,34,35] and reflects the differences between known, frequent words and completely unknown words at early stages of their lexical processing, which may be related to the process of whole-form lexical access. This advantage exhibited by familiar words in their orthographic processing and in their whole-form access, as compared to unknown words, is best explained by the activation of preexisting memory traces for familiar linguistic items. This advantage is also reflected in the pattern of results obtained at source level, in which known words showed higher activation than unknown words at left frontotemporal brain areas, typically related with language processing.

## 4. Experiment 2: Phonological Training

This experiment was aimed to determine whether novel word training in spoken domain would cause a facilitation in the visual recognition of the respective orthographic stimuli at early stages of their reading, underpinned by the activation of previously trained phonological codes. Taking into account lexical differences observed in P200 component in Experiment 1, it was expected that such putative facilitation would affect the amplitude of this ERP, leading to reduced P200 differences between known and previously trained words. Moreover, a modulation of the LPC component was also expected due to repeated exposure to novel word-forms. That pattern of results would for the first time provide neurophysiological evidence of cross-modal top-down facilitation processes during visual recognition of novel written word-forms.

### 4.1. Method

**Participants**. A group of twenty-two participants (three males; mean of age = 22.40, SD = 2.32), who did not take part in Experiment 1, took part in this experiment for course credit. All of them gave their consent to participate. Criteria for the selection of these participants was the same as in the previous experiment.

**Materials**. A new set of materials was selected in order to carry out a phonological training for the novel words. Thus, utterances for each of the 24 unknown words, produced by a female, Spanish native speaker, were recorded for its presentation to participants during the *training phase* of the experiment. Utterances were processed in Praat software [60] for accuracy check and necessary latency adjustment. The duration of recordings was approximately the same for all words (mean = 724 ms; SD = 81.94). Regarding the *testing phase* of the experiment, the same set of materials presented in Experiment 1 was used.

**Procedure**. The experiment consisted of two different phases, a training phase and a testing phase (see Figure 1). First, participants underwent a *training phase*. This was, explicitly introduced as a training to learn novel words; participants were required to listen as attentively as possible to the words presented and to repeat aloud each word after its auditory presentation. The purpose of repetition was to increase the phonological knowledge about the trained words by means of overt articulation. The novel spoken word-forms were presented to participants through headphones across 6 different training blocks (i.e., each novel word was presented six times) by means of E-prime 2.0 software (Psychology Software Tools Inc., Sharpsburg, PA, USA). The order of presentation was randomized within each of 6 training blocks. See Figure 1A for the sequence of presentation used during the phonological training.

Immediately after the training phase, an EEG cap was mounted on the head of participants and the next *testing phase* started (see Figure 1B). Thus, the lapse between the end of the training and the beginning of the test phase was approximately 45 min. During the testing phase, participants carried out the same silent reading task as used in Experiment 1, in which the orthographic form of the previously trained stimuli was presented together with control known words (see Figure 1B). The task was introduced as a silent reading task, in the same manner as in the Experiment 1.

**EEG recordings and analysis**. Recording and preprocessing of EEG data was carried out in in the same way as described in Experiment 1. The same procedures were followed for cleaning ERP data, including ICA for the removal of ocular artifacts (average number of rejected ICA components: 2, range = 0–7; average number of interpolated channels: 4, range = 1–8), triangular interpolation for bad channels (mean number of interpolated channels: 5, range = 1–10) and additional artifact rejection for removal of any remaining contaminated epochs (mean number of rejected trials per condition was, for known words: 4, range = 1–6, representing 18.18% of data; for novel words: 3, range = 0–6, 13.26% of data). ERP analysis for the silent reading task were carried out as described in Experiment 1 (the average number of included trials per condition: known words: 20, range = 18–23; novel words: 21, range = 18–24).

### 4.2. Experiment 2 Results

Results from the permutation *t*-test showed significant differences (*p* < 0.025) between known words and those word-forms previously trained in phonology at a late time window, ranging approximately from 520–780 ms, broadly distributed at central and posterior scalp regions and maximal at central sites. In particular, this result showed that trained words exhibited more positive amplitudes than known words at this time range. Importantly, no other differences were found along the whole ERP segment (see Appendix A). Both the inspection of ERP waveforms and the topographic distribution of this effect likely suggest the modulation of the LPC component, with higher enhancement of the LPC amplitude for the newly-trained words at spoken domain than for already known words. See Figure 3A.

At source level, differences in current density map between both conditions at the 520–780 ms averaged time window revealed that trained words showed higher activation than known words in the right superior occipital pole and the left middle anterior pole; thus, the higher LPC enhancement obtained at surface level for trained words is likely generated by these neural sources. On the other hand, known higher exhibited higher activation than newly-trained words in right frontotemporal areas, including the inferior frontal gyrus, right precentral and postcentral gyri and supramarginal gyrus, as well as at bilateral occipital regions. See Figure 3B.

### 4.3. Experiment 2 Discussion

An LPC lexicality effect was found in experiment 2, reflecting post-lexical differences between the processing of known and newly trained word-forms. The larger LPC responses exhibited by novel written words in comparison to non-trained familiar words are likely a consequence of their repeated exposure during the training phase, as found in previous studies which used repetitive presentation of novel word-forms [41,46,47]. Therefore, such an LPC lexicality effect, together with the higher activation of the left temporal pole exhibited by trained words likely reflect the access to episodic memory traces for these stimuli, built-up across their phonological repetition.

Moreover, no P200 lexicality effect was found in this task, reflecting similar orthographic processing between known words and those novel written words previously trained at spoken domain. Therefore, the pattern of results obtained in P200 and LPC components across both experiments suggest the cross-level top-down facilitation at both lexical and post-lexical stages during the processing of novel written words, as a consequence of their previous phonological training.

## 5. Experiment 3: Combined Phonological and Semantic Training

In this experiment, the putative additional facilitation in the visual recognition of novel written word-forms lead by top-down access to semantic information was studied by training these stimuli both in their phonological form and meaning. Such additional training was expected to promote the recollection of semantic cues during the visual presentation of novel written word-forms at the reading task and thus facilitate their lexico-semantic processing, likely reflected in the modulation of the semantically-related N400 component.

### 5.1. Method

**Participants**. Another group of twenty-two undergraduate students took part in the experiment for course credits (two males; mean age = 20.63, SD = 2.51). All of them gave their consent to take part in the experiment. These participants were different from those in the previous two experiments. Their selection criteria were the same as in Experiments 1 and 2.

**Materials**. A new set of materials was selected in order to carry out the additional semantic training for the novel words. In particular, a set of 24 photographs was selected, each of them to be presented in association with each particular spoken word-form during the training phase (i.e., a photograph of the jujube fruit, presented together with the novel word *jínjol,* see Figure 1A). All of them were color photographs with similar size and appearance (520 × 360 cm on average). The spoken word-forms were the same as presented in Experiment 2. For the testing phase, the same materials were used as in Experiments 1 and 2.

**Procedure**. Similarly to Experiment 2, this experiment consisted of two phases, training and testing. During the *training phase*, participants were introduced with the novel word learning task. They were required to listen as attentively as possible to the words presented, paying attention to the photograph displayed in the screen, and to repeat each word aloud after its auditory presentation. The procedure was the same as carried out in Experiment 2, with utterances for novel words repeatedly presented to participants through headphones. However, in this experiment, the phonological form of the unknown words was presented together with a photograph in order to present the word´s meaning. The photograph was displayed in the middle of the screen over a black background for 1700 ms, followed by the audio playback starting at 700 ms after the photograph onset (see Figure 1A). A total of 24 word + photograph pairs were presented across the six different blocks of repetition. The same associations between phonological word-forms and photographs were carried out across the training phrase, with the presentation of each associated phonological form—photograph randomized within each block of exposure. Participants

After the training, participants were prepared for EEG recordings and they underwent the *testing phase*; during this phase, participants carried out a silent reading task, with the presentation of the orthographic form of previously trained novel words together with control known words (see Figure 1B). The procedure for this task was the same as described in Experiments 1 and 2.

**EEG recordings and analysis**. Recording and preprocessing of EEG signals was carried out in the same manner as described in previous experiments. Cleaning ERP data entailed the same steps, including ICA for the removal of ocular artifacts (average number of rejected ICA components: 2, range = 0–5; average number of interpolated channels: 4, range = 1–8), triangular interpolation for bad channels (mean number of interpolated channels: 4, range = 1–9) and additional artifact rejection (mean number of rejected trials per condition was, for known words: 4, range = 1–6, representing 18.18% of data; for novel words: 3, range = 0–6, 14.20% of data). ERP analysis for the silent reading task were carried in identical way as described in Experiment 1 and 2 (the average number of included trials per condition: known words: 20, range = 18–23; novel words: 21, range = 18–24).

### 5.2. Experiment 3 Results

Permutation *t*-test revealed reliable differences (*p* < 0.025) between brain activations exhibited by known words and novel written words previously trained in phonology and meaning at a period ranging approximately between 300–500 ms post-stimulus onset; no other differences were found along the entire ERP segment (see Appendix A). These differences were broadly distributed across fronto-central and posterior regions, maximal at posterior scalp sites and revealed that newly-trained words elicited less negative amplitudes than known words. Both the latency and topographic distribution of this effect as well as the inspection of the ERP waveforms suggested the modulation of an N400-like component (see Figure 4A). Thus, combined training at both phonological and semantic levels produced a reduced activation for newly-trained words in comparison to well-known but non-trained words in a time window coinciding with the N400 component.

The estimation of neural sources for both conditions at the 300–500 ms averaged time window revealed that novel written words previously trained on both phonology and meaning exhibited higher activation than known words in the left angular gyrus, whereas known words showed higher activation than trained words in the left and right occipital poles. See Figure 4B.

### 5.3. Experiment 3 Discussion

Experiment 3 revealed that novel written words trained in both their phonology and their meaning exhibited reduced negative responses in comparison to the known words at a time window coinciding with the N400 component. Importantly, the effect of this training was constrained to this N400-related time window, with no modulation found either at earlier or later time windows related to the P200 or LPC components, respectively. Such an N400-like reduction could reflect a lexico-semantic facilitation, in particular the ability of participants to recover information related to a semantic referent for the novel words, trained immediately before the testing phase; in agreement with this idea, the reading of these stimuli recruited the left angular gyrus at this particular time window. These findings contrast with the results found for known words, which importantly were not repeatedly associated to any semantic reference in the context of the experiment. Therefore, this pattern of results suggests the interplay of cross-level top-down facilitation during the visual recognition of novel written words, particularly at a late, lexico-semantic stage.

## 6. General Discussion

The present study aimed at investigating the neural dynamics underpinning the putative cross-level top-down facilitation during the processing of novel written word-forms, as a result of previous experience at different levels of representation. To address this question, an EEG was recorded during the first visual encounter with novel written word-forms, which had been pre-trained in either their phonological form only or both in the phonological form and associated meaning. Familiar words were also presented at this reading task, serving as control stimuli with pre-existing fully formed memory traces. Importantly, both types of stimuli were meticulously controlled, ensuring that the differences between trained and known words obtained across the experiments could only being explained by the effect of the trainings applied. Our findings reflect a dissociated pattern of neural responses depending on previous exposure, with phonological experience causing a modulation at a late, likely post lexical stage of visual word recognition (~520 ms), and with additional semantic training affecting a stage related to lexico-semantic processing (from ~300 ms onwards). Importantly, the early lexical differences (~180 ms) detected during the reading of known and untrained stimuli did not emerge when novel words received a previous training at spoken domain. Therefore, these results may reflect the cross-level top-down interactive effects during the acquisition process of novel vocabulary. In what follows, the effects found are discussed in more detail.

The results found in Experiment 1 revealed a stronger P200 response for familiar in comparison to novel written word-forms, likely produced by the activation of established memory representations for known stimuli. Consistent with previous findings, such a P200 lexicality effect found in Experiment 1 likely reflects differences between familiar and unknown stimuli during the extraction of visual features at early stages of visual word recognition (namely, grapheme-to-phoneme decoding), with larger amplitudes possibly reflecting holistic recognition of the word-form for familiar words [31,32,33,34,35]. Moreover, our source estimation data, although complementary, are also compatible with this interpretation, with two language-related areas within the left perisylvian cortex (left inferior frontal gyrus and left temporal gyrus) as the most likely neural sources for the P200 responses elicited by familiar, known words at the surface level. In this line, the early activation of the left inferior frontal gyrus, higher for known words than for letter strings, has been related to grapheme-phoneme decoding processing and the assembly of phonological codes in reading [61,62,63]. Besides this, the higher activation found at the left middle and superior temporal lobe for familiar words in comparison to novel written word-forms likely reflects the early activation of semantic representations for known stimuli, in agreement with in previous ERP data using source estimation (i.e., [64]).

However, when novel written word-forms received previous phonological training, either with or without semantic information (Experiments 2 and 3), these stimuli produced early brain responses indistinguishable from those exhibited by familiar words. Taking into account interactive models of visual [7,8,9] and spoken [10,11] word recognition, the efficiency during this process is ensured by a continuous and bi-directional interaction between the sensory input (i.e., novel written word-form) and the available information at other levels of representation (i.e., phonology, semantic). In this sense, the absence of a P200 lexicality effect in Experiments 2 and 3 likely suggest the top-down activation of phonological codes, established for these stimuli as a consequence of the previous experience, which in turn contribute to the parallel activation of grapheme-to-phoneme decoding processes during visual word recognition. Following such interactive approaches, two different facilitation mechanisms, driven by bimodal grapheme-to-phoneme (and thus phoneme-to-grapheme) decoding processes, could be responsible for the resemblance in orthographic processing between novel (but phonologically/semantically trained) written words and already lexicalized words.

On one hand, the phonological codes for novel words activated and stored during the previous training were likely re-activated during their grapheme-to-phoneme decoding at the visual domain. Thus, such activation of phonological information led to a top-down facilitation in the processing of trained words, which might enable their parallel grapheme-to-phoneme decoding in a similar fashion as for familiar words. In contrast, when phonological features were not been previously experienced (as in Experiment 1), the orthographic processing for novel and familiar words was significantly different, as reflected in the P200 amplitude, likely resembling the sublexical processing carried out for non-trained stimuli. On the other hand, the repeated exposure to novel words in the spoken domain likely contributed to the formation of new orthographic traces, even in the absence of visual experience. The fact that orthographic codes are activated during spoken language processing has been demonstrated in different studies, both at the behavioral [65,66] and, importantly, the neurophysiological level [67,68,69,70]. For instance, using an auditory lexical decision task, Perre and Ziegler [70] found differences in the recognition of spoken words with different orthographic consistency (namely, with consistent or inconsistent mapping of phonemes onto graphemes) at both early and late ERP components, thus indicating on-line activation of orthography during spoken word recognition. Although the present study did not track the neural activation during the training at spoken domain, the differential orthographic processing elicited by novel words depending on their previous training likely suggests that auditory exposure promoted the activation of orthographic patterns during this phase. Literacy exposure might be the reason for such an activation of orthographic patterns during the processing of spoken language. Indeed, during the process of literacy acquisition, there is a continuous association between orthographic and phonological features of words, with the activation of corresponding brain areas, such as fusiform and frontal gyri [30,71]; later on, when the reader has become competent, the presentation of phonological word-forms propagates the activation across the entire neural network built through reading experience, thus causing the processing of orthographic aspects of such phonological word-form even in the absence of visual input. Such an activation of orthographic features during phonological training likely led to building up the word´s representation also at the orthographic lexicon, which in turn contributed to the parallel, whole-word processing of these stimuli during their first visual encounter.

Therefore, our results reflect that in a consistent and completely transparent orthographic system such as Spanish, vocabulary exposure in the spoken domain seems to be highly beneficial for reading. Importantly, this effect could be also generalized to many other languages characterized by a shallow orthographic system, such as Finish, Italian, or Greek. Nonetheless, a different pattern of results should be expected under inconsistent orthographies (i.e., English), in which the lower reliability of phonology would lead to a slower representation of orthographic features through such bimodal decoding processing. Although behavioral studies have reported results consistent with this idea (i.e., [5]), more cross-linguistic ERP research is needed to further clarify whether inconsistent orthographies are less benefited by lower phonological top-down processes.

Furthermore, the results found in Experiment 2 indicate that the exposure with the phonological form of novel words also caused a later, post-lexical effect during their visual processing (~520–780 ms), compatible with the modulation of the LPC. LPC enhancements, resulting from repeated exposure to novel stimuli, have been considered as an index for the formation and strengthening of episodic memory traces for newly trained stimuli, which likely facilitates their recognition through episodic memory recollection [40,41,46,47]. Therefore, the LPC lexicality effect found in the present study likely reflects the access to episodic memory traces for phonological forms built up during the training phase, which does not occur in the absence of previous training. Moreover, the finding of the left anterior middle temporal cortex as one of most likely neural generators for the LPC activity showed by trained words is compatible with the recollection of phonological word-forms carried out during the reading of these stimuli. Indeed, this is a region mainly involved in memory retrieval processes [72,73,74], and it has been also suggested as a neural generator for LPC effects in previous ERP studies [47,75]. As a side note, no LPC lexicality effect was found either in Experiment 1 nor in Experiment 3. The lack of an LPC modulation in the Experiment 1 is not surprising, as the stimuli were not previously repeated. Regarding Experiment 3, although the trained words showed a late enhancement of their activity, that may be an after-effect driven by the N400 modulation, rather than an LPC modulation itself. The absence of an LPC modulation in Experiment 3, in which novel spoken word-forms were also repeated in association with a semantic referent, may be actually explained by the access to semantic information during such combined training, which likely promotes stronger and more interactive memory traces for these stimuli; this information, interconnected across both levels of representation, must be easily accessed during visual word recognition, resulting in less recruitment of episodic memory processes to assist in recognition. This pattern of results is compatible with that found in behavioral studies, in which the more information is provided during training (phonological and semantic rather than only phonological), the better the outcomes during visual word recognition are (i.e., [22]).

Besides phonological knowledge, the findings obtained in Experiment 3 indicate that previous experience at the semantic level influences the visual recognition of novel written word-forms at late lexical stages (starting around 300 ms), suggesting the involvement of semantic top-down facilitation mechanisms. Thus, novel written word-forms previously trained at both phonological and semantic level showed a reduced N400 component in comparison to familiar words; such N400- lexicality effect likely reflects a facilitation in the lexico-semantic processing of novel written words, triggered by the access of their semantic referent, represented by the photograph previously associated with their spoken form. Consistent with this idea, the left angular gyrus was found as the most likely neural generator of the N400 activity exhibited by newly trained words; this is an area typically related to multimodal concept representation and the retrieval of semantic memories [76,77,78,79], and is particularly involved in lexical access during visual word recognition [80]. Thus, the present finding supports the notion that meaningful training with novel spoken word-forms leads to access to recently acquired semantic memory traces for these stimuli during their reading (most likely activating the visual information extracted from the photograph), facilitating their lexico-semantic processing. Importantly, this pattern of results appears to be predominantly produced by a local repetition effect within this particular study, rather than reflecting general differences in the semantic processing of novel and known words. In this sense, although the known words were familiar and meaningful, these stimuli were not repeatedly associated with their semantic referents in a previous phase, and hence no semantically related modulation was observed. This idea is also supported by the lack of an N400 effect in both Experiments 1 and 2, in which the stimuli were not repeatedly associated with a semantic reference.

Finally, a cautious note should be given regarding our results across the three experiments. Taking into account that each experiment was carried out in a different group of participants, any comparison of the effects across experiments should be taken cautiously, even when our samples were selected from the same population and matched in age and educational and socioeconomic background. Indeed, potential differences not controlled between groups (i.e., experience in learning other languages) could explain the different impact that the training had across the experimental groups. However, it must be noted that both training groups showed a similar facilitation during the orthographic processing of novel written-word-forms, as reflected in the absence of P200 lexicality effects; hence, no sign of an advantage of one group over the other was observed for the learning of these stimuli, but rather specific LPC/N400 effects derived from the phonological or meaningful nature of the training carried out. Furthermore, only the group that had no previous experience with the novel words exhibited, as expected, a poorer lexical processing of these stimuli; therefore, it seems reasonable to assume that the training, rather than individual variability, was the most probable cause of the differences found across groups. Nonetheless, future research in novel word learning should account for the limitation of using different samples, testing the effect of different training regimens within the same sample as well as assessing the influence of different background factors in this process (experience with other languages, reading proficiency, general language skills, etc.). Besides that, the use of bigger sample sizes, as tested here (*n* = 22), as well as the inclusion of additional control conditions (i.e., unknown, non-previously trained words) would provide stronger evidence for cross-modal interplay during the acquisition of new vocabulary.

To conclude, the present study provides new evidence about cross-level top-down influences during the visual processing of novel word-forms. Our results suggest that previous experience with novel written word-forms in the spoken domain causes a facilitation during the visual processing of these stimuli, reflected at both early lexical and post-lexical stages of their processing. Moreover, although access to semantic cues does not seem to make an additional contribution to the early stages of visual recognition, it does promote facilitation during the lexico-semantic processing of novel written word-forms. Importantly, our results point that tools with extremely precise temporal resolution such as EEG appear ideal for determining the temporal dynamics of such cross-modal top-down facilitatory processes, and should be considered as primary methods in future studies within this strand of research.

## Figures and Tables

**Figure 1 brainsci-10-00717-f001:**
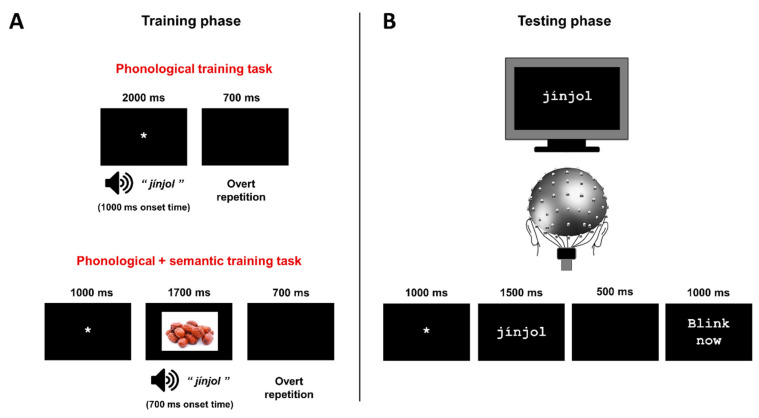
(**A**) Sequence of presentation during the training phase. For the phonological training (experiment 2), a fixation cross was presented in the center of the screen for 2000 ms; novel words were auditorily presented 1000 ms after the presentation of the fixation cross. A blank screen was presented for 700 ms, during which the participants pronounced the word presented immediately before. For the phonological and semantic training (experiment 3), the sequence of presentation was similar, except that, in addition to the sounds, a photograph of the corresponding object was displayed on the screen: a fixation cross was presented at the center of the screen for 1000 ms, followed by the presentation of a photograph, displayed for 1700 ms. An audio with the novel word was played 700 ms after the presentation of the photograph. Finally, a blank screen was displayed for 700 ms for the overt repetition of the word. (**B**) Sequence of presentation at the testing phase (Experiments 1, 2, and 3), during which the participants silently read the stimuli presented on the screen while their EEG signals were recorded. The sequence started with a fixation cross in the middle of the screen presented for 1000 ms and was followed by the target word, displayed for 1500 ms; a blank screen was then presented for 500 ms, and finally the message ‘’blink now’’ appeared for 1000 ms. Note that, for experiment 1, the participant underwent this phase directly with no previous training, whereas in experiments 2 and 3, the participants underwent the corresponding training phase (phonological or phonological and semantic, respectively) before the testing phase.

**Figure 2 brainsci-10-00717-f002:**
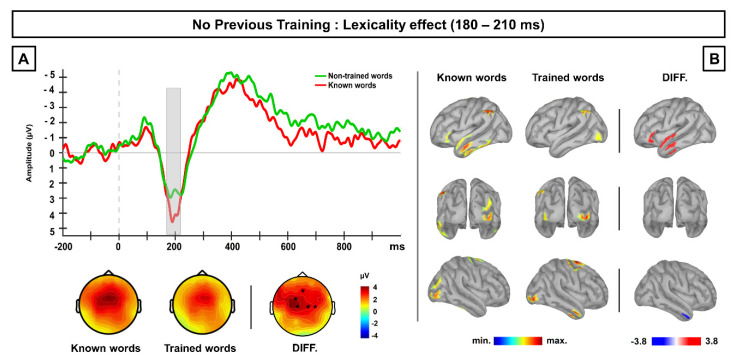
(**A**) Grand-average ERP waveforms at frontocentral scalp sites for the known and novel, untrained word-forms obtained in Experiment 1 (no previous training). Grey-shaded area highlights the time window (180–210 ms) at which the known words showed significantly larger neural responses than novel words, an effect compatible with the modulation of the P200 component. Topographic maps represent the ERP activity for each condition, as well the scalp distribution of the difference between them. (**B**) Current density maps obtained for each condition, as well as for the difference between both of them at the averaged time window, showing differences at the scalp level. Neural sources of the difference between conditions revealed the left inferior frontal gyrus and left temporal gyrus as the most likely neural generator for the larger P200 responses elicited by familiar, known words at the sensor level.

**Figure 3 brainsci-10-00717-f003:**
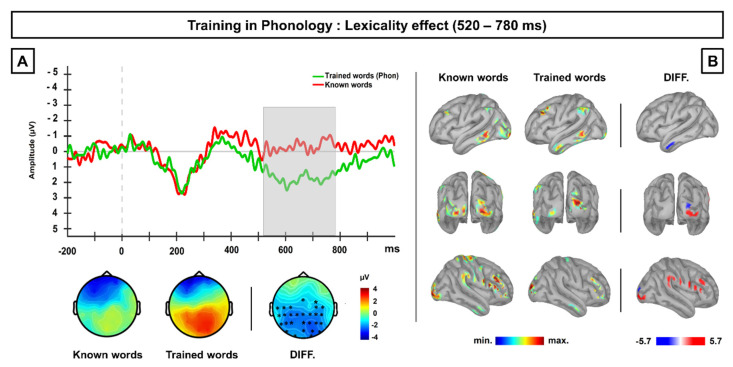
(**A**) Grand-average ERP waveforms at centro-posterior scalp sites for known and novel written word-forms after previous phonological training. Grey shaded area highlights the time window (520–780 ms) at which differences between conditions were found significant, with a like-LPC enhancement for previously trained novel written word-forms. Topographic maps displayed below represent the ERP activity for each condition as well the scalp distribution of the difference between them. (**B**) Current density maps obtained for each condition as well as for the difference between both of them over the time window showing significant differences at scalp level. Contrasts between conditions revealed the middle part of the left temporal pole as the most probable neural generator for the LPC enhancement found at surface for trained words.

**Figure 4 brainsci-10-00717-f004:**
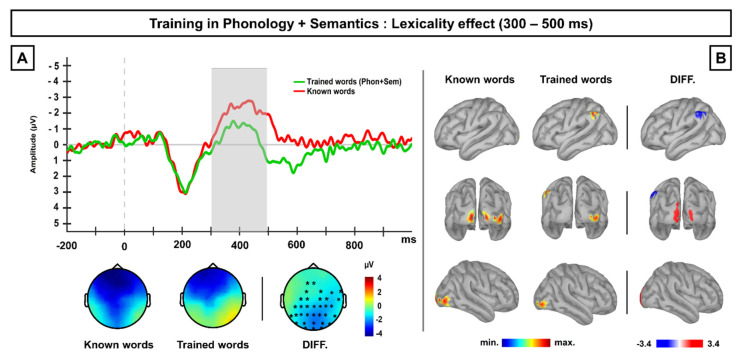
(**A**) Grand-average ERP waveforms at posterior scalp sites for known words and novel written word-forms previously trained at phonology and meaning. The grey shaded area shows the time window at which differences between conditions were found significant, corresponding with an N400 latency. Topographic maps displayed below represent the ERP activity for each condition as well the scalp distribution of the difference between them. (**B**) Current density maps obtained for each condition as well as for the difference between both of them at the averaged time window at which differences were found at scalp level (300–500 ms). Contrasts between conditions revealed the left angular gyrus as the most probably neural generator for the N400-like reduction found for trained word-forms at the scalp surface.

**Table 1 brainsci-10-00717-t001:** Main psycholinguistic properties of the stimuli used in the three experiments of the study. These were maximally matched between the experimental conditions. Standard deviation is shown in brackets. Independent-samples *t* tests confirmed no differences between the novel and known words across the variables.

	Novel Words	Known Words	*t* (46) Value	*p* Value
Lexical frequency	0	57.78 (103.99)	–	–
Number of syllables	2 (0)	2 (0)	0	1
Number of letters	5.50 (0.51)	5.50 (0.51)	0	1
Number of orthographic neighbors	1.42 (1.31)	1.46 (1.21)	−0.11	0.91
Bigram frequency (token type)	518.92 (285.91)	601.7 (350.51)	−0.89	0.37
Mean (1st and 2nd) syllable Frequency	2046.83 (3150.97)	2108.54 (2997.74)	−0.07	0.94

Novel words used in the study: cofín, dorna, fudre, bruño, gelfe, nabla, notro, pajel, paila, sisón, cuatí, facón, dolmán, puntel, reitre, roblón, runcho, seisén, holmio, trujal, jínjol, pambil, timple, carmes; Known words: color, toldo, valle, traje, golfo, bicho, litro, papel, nieve, mujer, baile, gafas, balcón, doctor, huella, millón, rastro, violín, garfio, crimen, cactus, césped, templo, pintor.

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
