# Peer review of "Neurophysiological Correlates of Top-Down Phonological and Semantic Influence during the Orthographic Processing of Novel Visual Word-Forms"

_brainsci, 2020, doi:10.3390/brainsci10100717_

Round 1

Reviewer 1 Report

The study uses three experiments to investigate the time-course of phonological and semantic top-down modulations by comparing novel and known words. The study involves a silent reading task (testing) followed or not by a training/exposure phase to novel words with either phonological or phonological and semantic exposure. The analyses involve both sensor-level and source-level analyses. Findings show increased P200 in Experiment 1 for known vs. novel words, indicating an early lexicality effect for known words. This is not seen in Experiment 2 nor 3, suggesting no differences after novel words have received exposure. Exp 2 and 3 however show later modulations for N400 and LPC. Complementary source-level analyses confirm the effects and their sources.

The study is ambitious in that it consists of three studies with sensor and source level analyses, though have small sample size. I do not see any major concerns about the methodology and overall narrative, however my main concern is regarding the analyses carried out.  I summarise my key points below:

Statistics

- At the sensor level, you carry out two statistical analyses: one to find the "time-electrode" clusters, and one to "define the topography". I understand that you want to use the second analysis to confirm the effect by averaging over the time window, but running two analyses on the same dataset is not advisable. You would be able to address the same question using cluster-based permutation tests, which protect you from multiple comparison problems (see http://www.fieldtriptoolbox.org/tutorial/cluster_permutation_timelock/). Otherwise, justify more clearly why these analyses are carried out in the way they are

- It seems that with your first analysis you are getting uncorrected p-maps to look for clusters. Are you correcting for multiple comparisons in the "second" round of t-tests? Some of the p-values are close to non-significance, and if these are uncorrected, they would not pass correction. As above, using cluster-based permutation tests would likely address these problems.

- I am assuming that you are looking at both ends of the tail (two-tailed tests i.e. you are looking at cond1 > cond2 and cond1 < cond 2). For this reason, you need to use a p threshold of .025.

> were source-level statistical analyses uncorrected? If so, justify why.

Exploratory analyses

> it is unclear why you are carrying out exploratory analyses; the rationale seem to be about confirming the effects, but is this necessary? It is reasonable to argue for a comparison of the effects across the three experiments, however it is not clear how this is being addressed by the analysis you have done.  You carry out post-hoc comparisons on the "Experiment" factor, to follow up on the interaction, but you have already looked at the ERP effect within each of the three experiments previously, so why isolate the levels in the experiment factor, and not look across experiments? Please clarify.
> I am concerned by the marginal significance of these values. Also, you state the lexicality effect is p = .1?
> In line starting 474 you state: the activity for novel and known words obtained in each experiment was estimated in the time window and scalp sites corresponding to each ERP of interest, according to the lexicality effect found in previous analyses in each experiment. It is not clear whether you are taking the time-electrode clusters from the first round of permuatation tests? Clarify what exactly was done here to feed into the ANOVA

Minor

> Proof read is needed as some parts are grammatically incorrect or difficult to read.
> for ease of readability, I suggest you tidy up the methods section. You could
provide a separate procedure section with the order of events that the participant experienced from beginning to end, including 297-201 (e.g. came into lab, received instructions, was capped up, started the task, finished the task). Then move the task description (187-191) in a separate sub-heading under the Materials (e.g. you can have Stimuli. Task). Also,
I would move the EEG information (192-196) to a separate section to integrate with the EEG analysis section, (e.g. EEG recording and analysis).
> specify that the vertex electrode is Cz (I assume?)
> did you measure horizontal eye movements/saccades? If not, was there a reason not to, e.g. given the reading task did you check where there were saccadic/horizontal eye movements?
> in the exploratory analysis you refer to Training and then Experiment as factors. You should be consistent

Author Response

Please see the attachment with a detailed response to all reviewer's comments.

Reviewer 2 Report

Please, find my review attached in PDF.

Author Response

Please, see the attachment with a detailed response to all reviewer's comments.

Reviewer 3 Report

This is an interesting study using ERP to investigate the neural dynamics underpinning what the authors refer to as "cross-level top-down facilitation" of processing novel written word-forms, with a manipulation of previous experience at different levels of representation. The paper is reasonably well written, although it has minor non-native turns of phrase, in particular with regard to definite/indefinite article presence/absence. This is a minor concern. Results are clear and this will be an interesting paper to the readers of Brain Sciences.

Author Response

We thank the reviewer for the time and effort evaluating our work and for the positive comments received. Regarding the minor concerned raised, please note that the whole manuscript has been proof read, in order to correct mistakes and to make our work more readable. Changes are highlighted thoughout the revised version of the manuscript.

Round 2

Reviewer 2 Report

I would like to thank the authors for having addressed my comments as well as the concerns raised by other reviewers. I am satisfied with authors' responses and would like to recommend this paper for publication. However, the revised version of the manuscript seems to contain a lot more English errors and formatting slips compared to the previous version. Please make sure that these are corrected before publication. Some examples below. Congratulations on the paper!

“This was , explicitly”
“Participants were seat in a comfortable chair”
“After place an EEG cap”

EXP 2: “average number of interpolated channels: 4, range=1-8), triangular interpolation for bad channels (mean number of interpolated channels: 5, range=1-10) “ - why is interpolation mentioned twice with different values?

“In particular, , a set of 24 photographs”

EXP 3: “average number of interpolated channels: 4, range=1-8), triangular interpolation for bad channels (mean number of interpolated channels: 4, range=1-9)” - same here, two different reports of interpolation

“the absence of a P200 lexicality effect in Experiments 2 and 3 likely suggest”